# Epigenetics and Gut Microbiota in the Pathogenesis and Treatment of Bipolar Disorder (BD)

**DOI:** 10.3390/cells14141104

**Published:** 2025-07-18

**Authors:** Shabnam Nohesara, Hamid Mostafavi Abdolmaleky, Ahmad Pirani, Sam Thiagalingam

**Affiliations:** 1Department of Medicine (Biomedical Genetics), Boston University Chobanian & Avedisian School of Medicine, Boston, MA 02118, USA; snohesar@bu.edu; 2Department of Medicine, Division of Gastroenterology, Beth Israel Deaconess Medical Center, Harvard Medical School, Boston, MA 02215, USA; 3Mental Health Research Center, Psychosocial Health Research Institute, Iran University of Medical Sciences, Tehran 1449614535, Iran; a.pirani85@gmail.com; 4Department of Pathology & Laboratory Medicine, Boston University Chobanian & Avedisian School of Medicine, Boston, MA 02118, USA

**Keywords:** bipolar disorder, epigenetic, microbiome, drug response

## Abstract

Bipolar disorder (BD) is a multifactorial mental disease with a prevalence of 1–5% in adults, caused by complex interactions between genetic and environmental factors. Environmental factors contribute to gene expression alterations through epigenetic mechanisms without changing the underlying DNA sequences. Interactions between the gut microbiota (GM) and diverse external factors, such as nutritional composition, may induce epigenetic alterations and increase susceptibility to BD. While epigenetic mechanisms are involved in both the pathogenesis of BD and drug treatment responses, epigenetic marks could be employed as predictors and indicators of drug response. This review highlights recent studies on the potential role of epigenetic aberrations in the development and progression of BD. Next, we focus on drug response-related alterations in the epigenetic landscape, including DNA methylation, histone modifications, and non-coding RNAs. Afterward, we delve into the potential roles of GM-induced epigenetic changes in the pathogenesis of BD and GM-based therapeutic strategies aimed at improving BD outcomes through epigenetic modifications. We also discuss how BD drugs may exert beneficial effects through modulation of the GM and the epigenome. Finally, we consider future research strategies that could address existing challenges.

## 1. Introduction

Bipolar disorder (BD), as a multifactorial disease in origin, is caused by interplays between genetic and environmental factors, with a prevalence of 1–5% in adults [1,2,3]. The main symptoms of BD are the recurrence of manic, hypomanic, and depressive episodes that may be accompanied by intervals of well-being [4]. BD is classified into two types: (i) BD type I (BD I), characterized by a manic episode with or without episodes of major depression, and (ii) BD type II (BD II), characterized by major depressive episodes and at least a hypomanic episode [5]. Given the elevated risk of suicide and disability, BD imposes a significant socioeconomic burden on both the family and society [6,7]. Owing to a remarkable link between heritable traits, epigenetic modifications, and drug response in patients with BD, a growing body of investigations suggests that genetic and epigenetic marks may have predictive value [8]. The explorations of reliable genetic and epigenetic markers of drug response pave the way for a deeper understanding of drug mechanisms of action and improved treatment management for individuals with BD [9]. Multiple lines of evidence over the last decade indicate that, besides genetic factors, epigenetic mechanisms, which regulate gene expression without changing DNA’s sequences, can serve as potential biomarkers for assessing both the risk of BD in vulnerable individuals and evaluate their response to various drug treatments [10].

The pathogenesis of BD and responses to medications are also related to diets and other environmental factors, such as gut microbiota (GM) [11]. A variety of environmental factors such as physical activity, diet, and plant extracts contribute to shaping the GM, and thus affect mental health involving the gut–brain axis [12,13,14,15].

For instance, in a large cohort study of subjects with psychiatric disorders (*N* = 1,101), including BD (*N* = 217) and controls (*N* = 343), an elevated risk of acute mania has been linked to the consumption of nitrated cured meats [16]. This study also demonstrated that consuming nitrated products in rats is associated with an elevated abundance of some gut bacteria, including *Lachnospiraceae* and *Erysipelotrichales*. Low consumption of polyunsaturated fatty acids (PUFAs), which maintain intestinal epithelium permeability, may give rise to developing BD as well [17,18]. On the other hand, anerobic exercises may support mental health by improving the diversity and abundance of genera from the *Firmcutes* phylum [19].

Therefore, physical activity, manipulating diet, and utilizing antimicrobials and drugs that affect brain function and cognition may provide opportunities to mitigate the development or exacerbation of BD by changing microbial communities and their metabolites, which affect epigenetic mechanisms [20,21]. Microbial communities in the human gastrointestinal tract may also influence responses to pharmacotherapy by affecting drug pharmacokinetics or manipulating the function of hepatic enzymes [22,23]. In fact, the GM, which includes a complex and dynamic population of pathogenic, commensal, and symbiotic microorganisms, possesses a bidirectional communication with the host, influencing their metabolic status [24,25]. Hence, numerous studies have focused on the potential roles of GM composition in the pathogenesis of BD and response to BD drugs. For example, Hu et al. reported that untreated BD patients exhibit a decreased abundance of butyrate-producing bacteria, which are involved in modulating both inflammation and the epigenome [26].

In this narrative review, we first discuss the relationship between epigenetic aberrations and BD pathogenesis. Then, we review the literature on drug response-induced alterations in epigenetic marks to evaluate whether epigenetic markers may be considered as reliable predictors of drug response. Next, we review the key role of the GM in developing BD and GMbased therapeutics for the treatment of BD patients via epigenetic mechanisms. We also discuss how BD drugs may exert their effects by modulating the GM through epigenetic changes.

## 2. Epigenetic Alterations and the Pathogenesis of BD

### 2.1. DNA Methylation

The main epigenetic mechanisms include DNA methylation (DNAm), post-translational histone modifications, and non-coding RNAs, whose interplay modulates and fine-tune gene expression levels [27,28]. Distinct DNAm patterns are initiated by the addition of a methyl group to a cytosine–guanine dinucleotide (CpG) across the genome and are generally associated with the suppression of gene expression. However, methylation of some particular CpGs in the gene promoter region or gene body may increase gene expression [29,30]. Genes’ transcription fine-tuning can occur mostly by DNAm or demethylation of CpG dinucleotides present in the genes’ promoter regions [31].

Aberrant DNAm of several genes’ CpG sites has been connected to BD and suicide. For example, Li et al. reported that BD patients exhibit aberrant DNAm in CpG islands (CGIs) flanking regions and CGI sparse promoters in blood cells [32]. Zhang et al. reported 2075 aberrant CGI methylation points in subjects with BD versus normal individuals, which were classified into 24 categories [33]. In their investigation, the authors found DNA hypermethylation of COMT and PPIEL genes in the blood samples of the observation group compared to the control subjects [33]. Gaine et al. found a noticeable hypomethylation of the ARHGEF38 region across four CpG sites in postmortem brain samples from BD patients who died by suicide versus healthy controls [34]. The main findings from other studies regarding the critical roles of DNA methylome alterations in individuals with BD are summarized in Table 1.

### 2.2. Histone Modifications

In addition to DNAm, the pathogenesis of BD is associated with changes in histone modifications [51]. Elimination of acetyl groups by histone deacetylases (HDACs) from lysine residues on the tails of histone proteins leads to condensation of the chromatin and a lower permissive state for transcription [52]. On the other hand, HDAC inhibitors (e.g., sodium valproate) provide a more active chromatin structure and subsequently increase genes’ transcription [53]. In vivo imaging of HDAC-specific radiotracers demonstrated that changes in the level and activity of HDACs are connected to attention and emotion regulation in subjects with BD [54]. Therefore, inhibitors of HDACs such as valproic acid (VPA) may be considered good candidates for exerting anti-manic effects by influencing specific regions of the brain [55].

### 2.3. Non-Coding RNAs

Among other epigenetic mechanisms, miRNAs and other non-coding RNAs also act as post-transcriptional regulators of gene expression [56]. In this field, an animal study reported that the development of manic-like behaviors after 72 h REM sleep deprivation (SD) in rats is correlated with reduced expression levels of miRNA-325-3p, miR-326-3p, and miR-330-5p in the prefrontal cortex (PFC) [57]. In BD, individuals with a higher genetic risk of developing the disease state exhibited increased expression levels of miR-15b, miR-132, and miR-652 [58]. Furthermore, it has been shown that depressive episodes of the BD subjects are connected to downregulation of miR-499, miR-708, and miR-1908 [59]. A study by Bame et al. identified 58 differentially expressed miRNAs between control- and BD patient-derived neurons, some of which are involved in axon guidance, cell migration, dendrite and synapse function, and brain development [60]. Considering these findings, it has been proposed that miRNA may be used as a potential biomarker for BD in the clinical settings and for developing new therapeutic agents and approaches in the near future [61].

In addition to miRNA, dysregulation of long non-coding RNAs (lncRNAs) is linked to BD pathogenesis. While a previous study reported reduced levels of the lncRNA MALAT1 in the blood cells of patients with BD [62], a more recent study has shown increased expression of two other lncRNAs, HOXA-AS2, and MEG3 in the blood samples of patients with BD [63]. Other findings related to the potential roles of miRNA dysregulation in BD pathogenesis are summarized in Table 2.

## 3. Epigenetic Alterations in Response to Mood Stabilizers and Antipsychotic Drugs

### 3.1. Valproate

VPA is a short-chain fatty acid, which is served as an FDA-approved mood stabilizer for the treatment of BD and an anticonvulsant drug for the epilepsy treatment [73,74]. VPA has demonstrated a capacity for changing the gene expression of cell cycle regulators via modifying promoter DNAm, which may influence cell proliferation in the hippocampus region [75]. VPA is also capable of inducing chromatin remodeling and attenuating DNA hypermethylation of GABAergic genes’ promoters that are responsible for disturbances in GABAergic neurotransmission in psychiatric diseases [76]. Dong et al. found that the protective impact of valproate in decreasing anxiety, cognitive dysfunction, and symptoms of BD is linked to elevated expression levels of reelin (RELN) gene and glutamic acid decarboxylase 67 (Gad-67) via decreasing DNAm of their promoters in nuclear extracts from the brain of adult mice [77].

Regarding DNAm, Aizawa et al. found that VPA administration could elevate the expression of p21 (both its mRNA and protein levels) in the mouse hippocampus, accompanied by changes in DNAm at the distal CGI of the p21 gene, without changing the expression of DNA methyltransferase (Dnmt) 1 or Dnmt3a [78]. Backlund et al. found that combination therapy with lithium (Li) and VPA in BD patients resulted in a hypermethylated pattern in leukocytes compared to Li monotherapy [79].

As another example, Chen et al. reported that prolonged (3 days) VPA treatment of mouse fibroblast 3T3-L1 cells could influence mitochondrial epigenetic marks by reducing 5-hydroxy methylation (5hmC) of mitochondrial DNA [80]. D’Addario et al. found that the increased DNAm of the BDNF promoter in subjects with BD II was reversed by various pharmacological therapies, especially VPA and Li [81]. D’Addario et al. also reported DNA hypermethylation at the PDYN gene promoter in BD IBD II subjects compared to healthy controls, which was reversed by both VPA and Li [44].

In addition to its effects on DNAm, VPA acts as a potent inhibitor of HDACs at therapeutic concentrations (0.4–0.8 mM), contributing to the regulation of gene expression through both HDAC-dependent and -independent mechanisms [82,83,84,85]. VPA has also been shown to activate the human FGF1 gene promoter by suppressing HDAC and GSK-3 activities [86]. Moreover, VPA is capable of inducing histone H3 methylation and acetylation in the leptin receptor gene in the hippocampus of Brown Norway rats [87]. In BD patients, treatment with VPA for four weeks increases the levels of acetylated Histone 3 (H3ac) and acetylated Histone 4 proteins (H4ac), as observed in lymphocyte nuclear protein extracts [88].

In animal models of acute mania, hyperactivity is linked to elevated HDAC activity in the PFC, and VPA has been shown to improve BD-like symptoms [89]. Valvassori et al. also developed an animal model of mania induced by d-AMPH and found elevated activity of HDACs in the PFC [90]. In their study, treatment of animals with VPA could reduce the activity of HDACs in the PFC and striatum of rats. In a rat model of mania induced by ouabain, Varela et al. found that VPA was able to reverse hyperactivity by reducing the activity of DNA methyltransferase and HDACs [91]. Logan et al. found that normalizing the manic-like behavioral profile in the ClockΔ19 model of bipolar mania in mice, treated with VPA, is mediated by HDAC2 inhibition in the ventral tegmental area (VTA) [92].

Besides DNAm and histone modifications, VPA has been shown to influence neuronal function by altering the expression of non-coding RNAs such as microRNAs [93,94]. Kim et al. found that chronic treatment with VPA is capable of influencing the expression of a BD-associated miRNA, miR-1908-5p, and its target genes such as *STX1A*, *CLSTN1*, *DLGAP4*, *GRIN1*, and *GRM4* [95]. In their studies, VPA reduced the expression of miR-1908-5p in neural progenitor cells derived from dermal fibroblasts of a BD patient while increasing its expression in control human neural progenitor cells. Additionally, it has been shown that lncRNAs may contribute to the mechanisms of action of BD drugs. For example, VPA could increase the expression of the hub lncRNA of module 5, GAS6-AS1, in human neuronal-like (NT2-N) cells [96]. Figure 1 illustrates a summary of findings related to the effects of VPA on different epigenetic mechanisms.

### 3.2. Lithium (Li)

Li is one of the gold standard drugs and first-line treatments for BD [97]. Approximately 20–30% of patients with BD are excellent responders to Li, while more than 40% show little to no clinical response [98]. Differential DNAm patterns have been identified between responders and non-responders to Li treatment. Claire et al. reported 111 differentially methylated regions (DMRs) in the DNA of blood cells between the responders and non-responders to long-term Li treatment [99]. Zafrilla-López et al. reported 130 differentially methylated positions (DMPs) and 16 DMRs between responders and non-responders to Li treatment, affecting 122 genes, including *CACNA1B*, *ANK3*, *TENM2*, *SORCS2*, *EEF2K*, *CYP1A1*, *HOXB6*, *HOXB3*, and *HOXB-AS3*, as well as genes involved in the GSK3β signaling pathway [100]. Furthermore, Asai et al. reported that CpG sites within the *SLC6A4* gene, which were hypermethylated in subjects with BD, exhibited significant hypomethylation in the neuroblastoma cells following treatment with Li, VPA, and carbamazepine [101].

Similarly to VPA, Li has also demonstrated the ability to reverse increased locomotor activity induced by d-AMPH in animal models of acute mania by inhibiting HDAC activity in the PFC [89]. In another animal model of mania caused by REM sleep deprivation, Andrabi et al. found that treatment with Li hampered derangements in histone acetyl transferase/histone deacetylase (HAT/HDAC) activity and suppressed inflammation, oxidative stress, mitochondrial impairment, and apoptotic cell death [102].

In humans, the beneficial effects of Li in BD have been shown to be mediated, in part, by the normalization of miRNAs’ expression. An earlier study demonstrated that gene-to-gene interplay between miRNA-206 and BDNF affects susceptibility to BD I and treatment response to mood stabilizers [103]. In a later study, Pisanu et al. found that miR-320a, miR-155-3p, and their targeted genes, CAPNS1 (Calpain Small Subunit 1) and RGS16 (Regulator of G Protein Signaling 16) for miR-320), and SP4 (Sp4 Transcription Factor) for miR-155-3p, may have a potential role in clinical response to Li [104]. In their study, increased levels of miR-320a and decreased expression of its target genes were observed in excellent responders versus non-responders [104]. In a large multinational GWAS study, miR-499a was also linked to Li response in BD [105]. In a more recent study, Cattane et al. reported 77 differentially expressed miRNAs in Li responders compared to non-responders. The affected miRNAs are involved in neurodevelopment, intracellular signal transduction, and immune system response [106]. Figure 2 provides a summary of the influence of Li on diverse epigenetic mechanisms and their corresponding genes in BD.

### 3.3. Lamotrigine (LTG)

Lamotrigine (LTG) is a phenyltriazine-derived anti-seizure medication and a mood stabilizer that is used for the treatment of depressive episodes in BD [107]. LTG, in combination with Li and VPA, is capable of alleviating the symptoms relevant to cyclic episodes of mania and hypomania, as well as depression [108]. LTG may exert its neuroprotective effects via epigenetic mechanisms. For example, Kidnapillai et al. found that LTG in combination with other BD drugs could elevate the expression of miR-128 and miR-378, which regulate the expression of genes involved in neurite outgrowth and neural plasticity [109]. Moreover, the protective effects of LTG may be associated with histone modifications. For example, the neuroprotective effects of LTG against glutamate excitotoxicity are linked to elevated levels of acetylated histone H3 and H4 and to increasing the histone acetylation and the Bcl-2 promoter activity [110].

### 3.4. Atypical Antipsychotics

Atypical antipsychotic drugs have been found to influence epigenetic mechanisms, especially DNA and histone methylation in neuropsychiatric disorders [111,112]. For example, quetiapine, as an atypical antipsychotic drug, is capable of reducing DNAm level of the promoter region of *SLC6A4* (serotonin transporter gene) in human neuroblastoma cells [113]. As another example, Houtepen et al. found that treatment of BD patients with both quetiapine and VPA alter their DNAm signatures in specific blood cells. In their study, the beneficial effects of quetiapine were related to immune and neurogenesis-related DNAm networks [114].

Non-coding RNAs are also considered as master modulators of the action of antipsychotic drugs in the treatment of BD patients. For instance, BD patients treated with a range of antipsychotics and mood stabilizers (Li, VPA, risperidone, olanzapine, clozapine, and quetiapine) for four weeks demonstrated higher plasma levels of miRNA-134 versus unmedicated BD patients [115]. Quetiapine and lamotrigine also increased the expression of miR-128 and miR-378 in cultured human neuronal cells, and their combination could reduce the expression of their target genes (*GRIN3A*, *VIM*, and *NOVA1*), which are involved in suppressing neurite outgrowth and neurogenesis [109]. Additionally, quetiapine is capable of elevating the expression of hub lncRNA LINC02381 in NT2-N cells [96].

The protective effects of asenapine and risperidone in subjects with BD are related to changes in miRNAs levels as well. Notably, 12 weeks of treatment with asenapine resulted in the overexpression of 14 miRNAs (miR-15a-5p, miR-17-3p, miR-17-5p, miR-18a-5p, miR-19b-3p, miR-20a-5p, miR-27a-3p, miR-30b-5p, miR-145-5p, miR-148b-3p, miR-106a-5p, miR-106b-5p, miR-210-3p, and miR-339-5p) and reduced expression of 2 miRNAs (miR-92b-5p and miR-1343-5p) in individuals with BD [116]. Furthermore, patients with BD treated with risperidone for 12 weeks exhibited downregulation of three miRNAs, including miR-146b-5p, miR-664b-5p, and miR-6778-5p [116].

## 4. The Gut Microbiota, BD, and the Disease Severity

Emerging evidence suggests that the GM is a key player in modulating mental health, while its composition changes are associated with the development of neuropsychiatric disorders such as BD [117,118]. Owing to the increased inflammation and disrupting intestinal barrier integrity, individuals with BD are affected by the translocation of bacteria and their toxins like lipopolysaccharide from the intestinal lumen to blood circulation and subsequently into the brain [119]. Disruption of the brain–gut-microbial axis in BD heavily affects host endocrine functions relevant to neuronal functions (i.e., controlling the production of neurotransmitters such as serotonin or mediating the activation of the hypothalamic–pituitary–adrenal axis) and host immune system in the intestine and brain [21].

The potential role of the GM in the pathogenesis of BD is related to epigenetic aberrations, since GM-derived metabolites, particularly SCFAs, can function as HDAC inhibitors. For example, a decreased abundance of *Faecalibacterium*, a butyrate-producing gut bacterium, was observed in subjects with BD, and its level was directly associated with self-reported health status [120]. Moreover, changes in the GM composition involved in the pathogenesis of BD are associated with altered DNA methylation patterns. For example, GM diversity in participants with BD was negatively correlated with methylation of the clock gene *ARNTL*, a main regulator of monoamine oxidase A transcription and circadian rhythms [121]. Also, the GM–miRNA interactions may play a key role in the pathogenesis of psychiatric disorders [122,123]. The GM influences host health by regulating host miRNAs. As an example, the inhibition of miR-34a-5p by an antagomir could improve total abdominal irradiation-mediated cognitive impairment through restoring GM composition [124]. The imbalances in gut bacteria and epigenetic metabolites, such as butyrate, in BD patients may also increase oxidative stress and neuroinflammation, which in turn negatively affect mood and cognitive function. To this end, Huang et al., investigated the alterations in the GM composition and its association with inflammation in depressed BD patients and reported decreased abundance of some bacteria (*Butyricicoccus*, *Lachnospiraceae incertae sedis*, and *Dorea*) involved in indirect release of anti-inflammatory cytokines such as IL-4, IL-10, and IL-11 [125]. In another study, untreated BD patients exhibited lower levels of the anti-inflammatory butyrate-producing bacteria versus healthy controls [26]. Painold et al. reported that the GM composition is associated with distinct changes in inflammatory and metabolic factors in BD patients. In their study, BD patients exhibited higher abundance of pro-inflammatory bacteria, like *Streptococcus*, which was linked to greater serum IL-6 levels [126]. Additionally, Coello et al. found that the pathogenesis of BD may be related to *Flavonifractor*, a bacterial genus that may contribute to inducing oxidative stress and inflammation in the host [127]. In a study conducted by Guo et al., unmedicated depressive BD II patients showed increased levels of pro-inflammatory bacteria (e.g., *Proteobacteria*, *Enterobacteriaceae*, *Porphyromonadaceae*, *Pseudescherichia*), higher levels of inflammatory cytokines levels, and changes in glutamate (Glu) and gamma-aminobutyric acid (GABA) metabolism-related bacteria (*Bifidobacterium*, *Escherichia*, *Bacteroides*, *Parabacteroides*, *Eggerthella*) [128].

## 5. Microbiota-Based Therapeutic Approaches for the Treatment of BD

Several approaches can be employed to modulate the GM and its associated epigenetic effects in managing BD.

### 5.1. Probiotics

Probiotics are microorganisms with the capability for creating health benefits to the host when they are consumed in proper concentrations [129,130,131]. Some probiotics, known as brain probiotics, can promote the production of neurotransmitters such as GABA and serotonin and improve brain functions [132,133]. Specific SCFA-producing probiotics known as “psychobiotics” have the potential to improve disease outcomes by mitigating epigenetic dysregulations, as many SCFAs are epigenetically active metabolites [134]. SCFA-producing probiotics like *Lactobacillus*, *Faecalibacterium*, and *Ruminococcus* can alleviate depressive-like behaviors and cognitive impairments in BD patients [135]. Similarly, probiotic organisms such as *Lactobacillus rhamnosus* strain GG and *Bifidobacterium animalis* subsp. *lactis* strain Bb12 have been shown to improve symptoms in patients with mania and BD Ⅰ, possibly through enhancing butyrate production, maintaining the mucosal barrier, and reducing oxidative stress and systemic inflammation [136,137,138,139]. Several other studies have shown that probiotic supplementation may improve manic symptoms and cognitive function in BD subjects [139,140,141]. As an interesting example, Zhang et al. found that oral administration of a probiotic containing *Bifidobacterium animals subsp. lactis BAMA-B06/BAu-B0111* could alleviate depressive mood in subjects with depressive BD type I by reducing the abundance of phylum *Firmicutes* and improving metabolic disarrays [137]. In another study, Borkent et al. found that beneficial effects of a probiotic, containing nine *Bifidobacterium* and *Lactococcus* bacterial strains in BD subjects, reduce intestinal permeability and suppress inflammation [142].

### 5.2. Postbiotics

SCFAs, well-known postbiotics and metabolic byproducts generated by the microbial community in the gut during the fermentation of dietary fibers, play an important role in host health, since they act as histone deacetylase inhibitors [143,144]. In addition to histone modifications, the beneficial effects of SCFAs, such as butyrate, in the treatment of neuropsychiatric diseases may also be attributed to their key role in modulating DNA methylation [145,146]. Several studies have demonstrated that SCFAs may be used to alleviate symptoms of BD in human and animal models [147,148]. For example, Moretti et al. found that rats exposed to an animal model of mania induced by d-AMPH exhibited hyperactivity and reduced activity of mitochondrial respiratory chain complexes. Treatment with sodium butyrate was able to counteract these abnormal changes [149]. Valvassori et al. suggested that sodium butyrate may be considered a potential new mood stabilizer, as it restores proteins from the neurotrophin family—including BDNF, NGF, and GDNF—and exerts neuroprotective effects against oxidative damage in the frontal cortex and hippocampus of rats exposed to an ouabain-induced animal model of mania [150,151].

### 5.3. Specific Diets (e.g., Ketogenic Diet)

Diet also has a great influence on improving the symptoms of subjects with BD through modulation of the GM composition and epigenetic modifications. For instance, a healthier diet containing fish, beans, fruits, and nuts triggers a greater diversity and richness of GM in BD patients [152].

Ketogenic diet (KD) is also a promising diet capable of mimicking the mechanisms of mood stabilizers for repairing or bypassing metabolism dysregulations, reducing neuroinflammation, and improving manic symptoms via two mechanisms [153,154]. First, KD influences the synthesis of neurotransmitters such as GABA and glutamate via increasing the production of acetyl-CoA from ketone bodies, particularly β-hyroxybutyrate (βOHB) [155]. Second, βOHB acts as class I HDAC inhibitor that elevates histone acetylation and thereby enhances the expression of several genes involved in mitochondrial function, antioxidant defense, and the reduction in inflammation [156]. βOHB also decreases HDAC2 levels, which in turn elevates the expression of genes involved in oxidative stress and neuroinflammation. Polito et al. found that βOHB produced by KD could increase microglial polarization towards an M2 anti-inflammatory phenotype, which in turn decreases the pro-inflammatory cytokine IL-17 and increases the anti-inflammatory cytokine IL-10 [157]. Sethi et al. reported that a ketogenic diet may contribute to improved metabolic health with a greater than 1-point improvement in 69% of subjects with BD [158].

### 5.4. Fecal Microbiota Transplantation (FMT)

Fecal microbiota transplantation (FMT) is an approach in which fecal bacteria or other microbes are transferred from a donor into the gastrointestinal tract of a recipient [159,160,161]. FMT may be considered a potential therapeutic procedure for investigating the role of gut microbiota in the pathogenesis of BD and its treatment in animal models [162,163]. For example, FMT from BD subjects to mice is associated with delivering behavioral and metabolic characteristics relevant to BD. In fact, mice receiving FMT from BD patients exhibited anxiety-like behavior and reduced sociability, which was associated with an elevated abundance of *Bacteroidota* and reduced abundances of *Parabacteroides merdae* and *Akkermansia muciniphila* and reductions in the levels of epigenetic metabolites, including acetic acid and butyric acid [164]. As another example, Hinton examined the effect of FMT in a BD I subject unresponsive to different types of psychotropic medications and found that this patient exhibited drug-free remission of psychiatric symptoms during the follow-up years [165].

### 5.5. Antipsychotic Medications

It is also worth noting that the atypical antipsychotics (AAPs) and non-AAPs strongly affect microbial communities and GM diversity in subjects with BD [166]. A recent study has demonstrated that BD patients treated with antipsychotics and Li exhibit an elevated abundance of 23 genera and 18 bacterial species, such as butyrate-producing bacteria (*Ruminococcus obeum*, *Eubacterium biforme*, and *Eubacterium rectale*) [167]. Kamath et al. found that lurasidone is also capable of modulating the GM via elevating the mean operational taxonomical units (OTUs) and alpha diversity in rats [168]. Treatment of BD patients with quetiapine (300 mg/d) for four weeks led to a rebound in the abundance of *Bifidobacteria* and *Eubacterium rectale*, bacteria involved in reducing inflammation by producing butyrate and other SCFAs [169].

Another recent study showed a positive relationship between lurasidone bioavailability, microbial diversity, and SCFA levels, which is mediated by adjusting luminal pH [170]. High concentrations of SCFA also provide a favorable environment for lurasidone solubility via reducing intestinal pH, whereas GM depletion leads to reduced solubility of lurasidone [170]. Notably, while lurasidone has not been associated with weight gain [171], there is evidence that adolescents treated with risperidone (versus psychiatric controls) exhibited a lower ratio of *Bacteroidetes*–*Firmicutes* bacteria, which contributed to weight gain [172]. Figure 3 illustrates how interactions between the GM and BD medications, mediated through epigenetic mechanisms, which contribute to therapeutic benefits for patients with BD.

## 6. Conclusions, Challenges, and Prospectives

The severity of BD and the outcome of therapy can be related to alterations in epigenetic mechanisms, including DNAm, post-translational histone modifications, and non-coding RNAs. Understanding epigenetic aberrations involved in the pathogenesis of BD and the mechanisms of action of BD drugs may help identify individuals susceptible to BD. This knowledge could also serve as a basis for developing biomarkers that enable more accurate diagnosis, treatment, and the exploration of novel personalized therapies. It could also pave the way for improving treatment outcomes in subjects with treatment-resistant BD who do not respond to available mood stabilizers and antipsychotic agents. In this review, we first examined the existing literature on the potential roles of epigenetic modifications in determining the therapeutic efficacy of mood stabilizers and antipsychotic drugs in BD patients.

One of the major limitations of these studies is that the vast body of current studies has not focused on the identification of epigenetic biomarkers in long-term therapies. Another challenge for pharmaco-epigenetic studies in BD is the small sample sizes and inadequate statistical power that should be addressed by using larger sample sizes and fostering collaborative cohort studies. Although multiple studies have shown that drugs such as VPA and Lithiumare are capable of targeting various epigenetic mechanisms, almost all published studies only examined one epigenetic mechanism at a time. Therefore, future studies should focus on the investigation of the impact of disease or a drug on a combination of several epigenetic mechanisms. Moreover, while GM research shows promise for developing more effective therapies to improve outcomes in BD patients, several significant gaps need to be addressed in future studies. For example, small sample sizes and the inconsistencies across studies, as well as variations in dietary habits and microbial composition across different populations, may lead to more controversial findings in the field. Additionally, most GM data are compositional, focusing on relative bacterial abundance, and therefore do not provide information on their specific absolute abundance or potential pathological impacts.

Replicability of GM research in BD could be more difficult since the disease has diverse features in different phases of BD. Additional difficulties come from technical differences (e.g., taxonomy databases, sequencing platforms, DNA extraction approaches, PCR primers) and investigating various patient cohorts (e.g., type1 or 2 BD, cyclothymia, rapid-cycling BD, BD with mixed features, BD with seasonal pattern, medicated vs. unmedicated, first-episode, chronic) that may cause systematic biases, which in turn hamper obtaining precise information about compositional differences in BD patients. Larger sample sizes and stronger demographic and phenotypic characterization of clinical samples in future studies could help to obtain greater insights into the functional roles of the GM in mental health and the pathogenesis of BD via epigenetic mechanisms. Likewise, other biological factors relevant to the gut–brain axis should be considered in investigations of the GM to facilitate better understanding of the mechanisms by which the gut ecosystem influences the cerebral tissue in BD patients.

Owing to the potential role of an accelerated biological aging and changes in biological clocks as well as epigenetic age dysregulation in BD, future studies should also focus on clarifying the correlation between age and gut microbial composition effects on epigenetic alterations. In order to determine whether alterations in the GM and gut-derived metabolites as epigenetic modifiers are trait- or state-related, more longitudinal sampling and studies should be implemented during different periods of treatment (e.g., acute vs. long-term), various stages and episodes (e.g., manic vs. depressive episodes) of BD. Despite current challenges, the prospects for novel discoveries in the GM–epigenome relationship and its influence on the brain through the “gut–brain axis” are incredibly promising. This excitement is largely due to recent technological advancements such as single cell sequencing, whole-genome epigenetic profiling, multi-omics approaches, large-scale multinational research initiatives, and advanced bioinformatics and machine learning.

## Figures and Tables

**Figure 1 cells-14-01104-f001:**
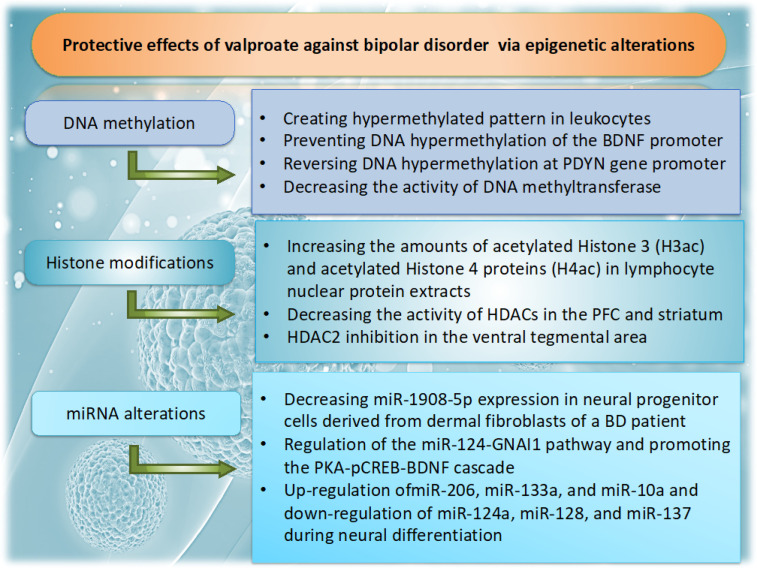
Protective effects of valproate mediated by epigenetic mechanisms in BD. Valproate is capable of exerting beneficial impact in BD patients via alleviating abnormal changes in DNA methylation, histone modifications, and miRNAs. Figure was created using the Microsoft PowerPoint software 2010.

**Figure 2 cells-14-01104-f002:**
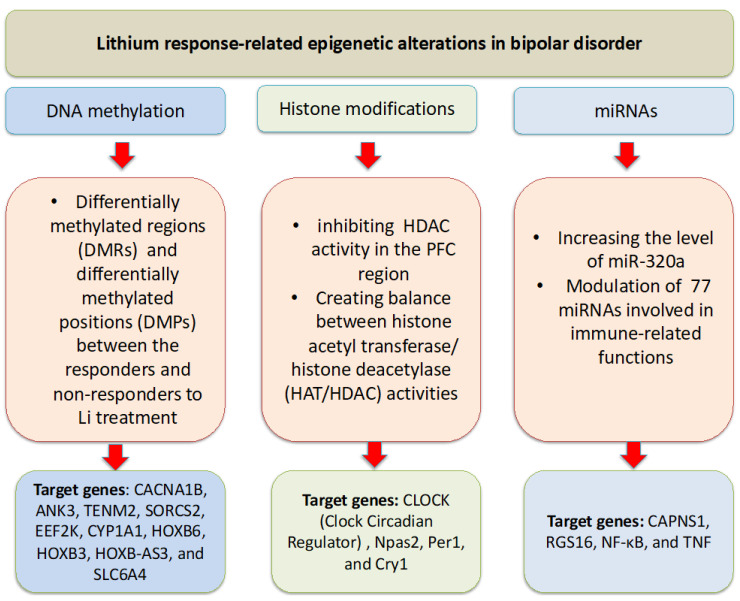
Lithium effects on different epigenetic mechanisms and their corresponding affected genes. Lithium has demonstrated the ability to exert its protective effects via normalizing aberrations in DNA methylation, histone modifications, and miRNAs. Figure was created using Microsoft PowerPoint software 2010.

**Figure 3 cells-14-01104-f003:**
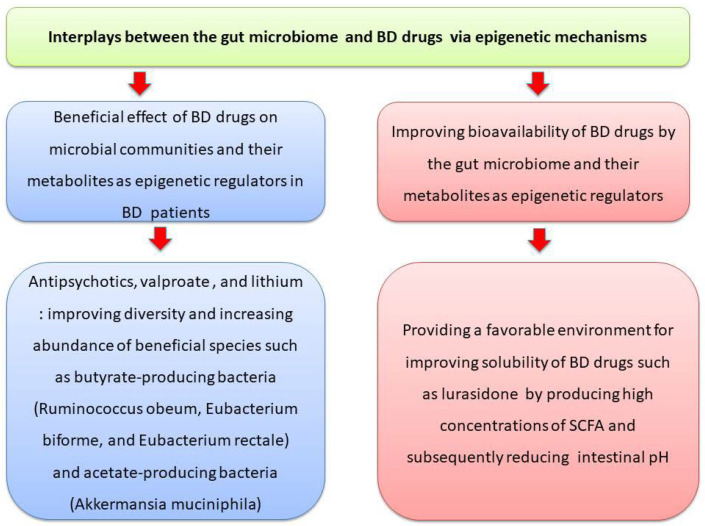
Interactions between the GM and bipolar disorder medications, which are mediated through epigenetic mechanisms and contribute to therapeutic benefits for patients with BD. Figure was created using the Microsoft PowerPoint software 2010.

**Table 1 cells-14-01104-t001:** Altered DNA methylation (DNAm) in BD patients versus control subjects.

Country of Studies/Number of Participants	Sample	Outcomes	Ref.
USA/115	Post-mortem brains (dorsolateral frontal cortex)	Hypomethylation of MB-COMT promoter and increased gene expression in BD, particularly in the left brain vs. controls	[35]
USA/35 patients with schizophrenia, 35 BD vs. 35 matched controls	Post-mortem brains (dorsolateral frontal cortex)	Hypermethylation of HTR2A promoter at and around the −1438A/G, but hypomethylation of its promoter at and around T102C polymorphic sites in BD vs. controls, which altered HTR2A expression	[36]
Canada/34 BD and 35 matched controls	Post-mortem brain, sperm, and blood samples	DNA hypomethylation at an extended HCG9 region (related to major histocompatibility complex and immune function) in BD subjects vs. controls	[37]
Italy/61 BD I and 50 BD II	Peripheral blood mononuclear cells	Greater amounts of BDNF promoter methylation in BD II vs. BD I, which, in general, decrease BDNF expression	[38]
Canada/16 controls, 14 BD with response to lithium, 16 unaffected relatives of BD subjects	Transformed lymphoblasts from subjects with BD	Reduced DNAm in BD patients vs. controls, which, in general, is linked to increased gene expression	[39]
USA and Iran/35 postmortem brains from BD and 35 from controls	Post-mortem brain samples	DTNBP1 promoter DNA hypermethylation decreasing gene expression in psychotic BD patients; DTNBP1 hypoactivity is linked to psychosis	[40]
USA/ 12 BD and 10 controls	Post-mortem prefrontal cortex	Lower KCNQ3 exon 11 DNAm and gene expression in BD vs. controls	[41]
UK/168 controls and 459 BD	Blood samples	DNA hypomethylation of FAM63B (a gene strongly linked to SCZ) in BD vs. controls	[42]
UK/ 40 BD subjects and 38 healthy controls	Blood samples	Complete methylation of four CpG islands (CGIs) across CACNA1C, a top gene linked to BD; hypermethylation of five CpG sites at CGI 3 island in intron 3 in BD vs. controls	[43]
Italy/99 BD 42 and controls	Peripheral blood mononuclear cells	Promoter DNA hypermethylation and reduced PDYN expression in BD IBD II vs. controls	[44]
USA/28 BD and 13 controls	Skeletal muscle samples	Higher global methylation of 5-mC and 5-fC in BD vs. controls	[45]
China/99 BD (N) and 92 controls	Blood samples	Hypermethylation of the AluY A1 and A2 CpG sites and hypo-methylation of A3 CpG site in BD	[46]
USA/166 BD and 162 controls	DNA samples from the Mayo Clinic Bipolar Disorder Biobank	Hypermethylation of CpG site created by Val = G allele of the Val66Met variance of BDNF, which decreases gene expression in BD associated with earlier disease onset	[47]
Japan/34 BD and 35 controls	Post-mortem brains (prefrontal cortex)	Hypermethylation of NTRK2 (encoding a BDNF receptor) and GRIN1 (encoding a subunit of the NMDA receptor) in neuronal cells of subjects with BD	[48]
USA/84 BD with and 79 without a history of suicide attempt vs. 76 controls	Blood samples	Six differentially methylated CpG sites and seven differentially methylated regions (DMRs) in BD with vs. without suicide attempt	[49]
USA/128 BD and 141 controls	Blood samples	Hypermethylation cg19215110 and cg23953820 sites but hypomethylation of cg14279856 and cg03270204 sites of DDR1 (encoding a tyrosine kinase receptor involved in neuronal migration) in BD	[50]

**Table 2 cells-14-01104-t002:** Dysregulated miRNAs in BD patients versus controls.

Country of Study/Number of Participants	Sample Source	Outcomes	Ref.
Italy/15 BD and 9 controls	Peripheral blood samples	Elevated expression of miR-150-5p, miR-25-3p, miR-451a, and miR-144-3p involved in metabolic processes but reduced expression of miR-363-3p, miR-4454 + miR-7975, miR-873-3p, miR-548al, miR-598-3p, miR-4443, miR-551a, and miR-6721-5p involved in neurogenesis and neurodevelopment in BD	[64]
Turkey/19 manic, 39 euthymic, and 51 controls	Blood samples	Up-regulation of miR-125b-5p, miR-106b-5p, miR-9-5p, miR-107, miR-29a-3p, miR-106a-5p, and miR-125a-3p in manic BD vs. controls and their association with manic episodes	[65]
Turkey/69 BD including 15 depressed, 27 manic, 27 euthymic and 41 controls	Plasma-derived exosomal miRNA	Reduced levels of miR-484, miR-652-3p, miR-142-3p, and an increased level of miR-185-5p in BD vs. controls, which are involved in the regulation of PI3K/Akt signaling, fatty acid biosynthesis/metabolism, extracellular matrix and adhesion pathways	[66]
Taiwan/79 BD IBD II and 95 controls	Blood samples	Up-regulation of miR-7-5p, miR-23b-3p, miR-142-3p, miR-221-5p, and miR-370-3p in BD IBD II vs. controls and their potential role as useful tools for the diagnosis of BD IBD II.	[67]
China/27 BD and 32 unipolar depressive disorder	Blood samples	Higher levels of miR-19b-3p in BD vs. unipolar depressive disorder that is involved in the pathway of inflammatory dysregulation associated with experiencing early childhood trauma	[68]
Germany/960 BD and 960 controls	Blood samples	Association between miR-499a dysregulation and BD progression and probable BD susceptibility	[69]
Lithuania/26 BD and 74 controls	Blood samples	Overexpression of let-7e-5p and miR-125a-5p in BD subjects vs. controls and their potential role as peripheral biomarkers for BD	[70]
Iran/240 controls and 260 BD	Blood samples	Increased expression of miR-23b-3p and reduced expression of miR-19b-3p in BD vs. controls and their potential role as biomarkers for BD	[71]
United Kingdom/112 independent samples (51 female, 61 male)	Human fetal brain tissue	Correlation between the elevated prenatal expression of miR-1908-5p and susceptibility to BD	[72]

## Data Availability

Not applicable.

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
