# Peer review of "Epigenetics and Gut Microbiota in the Pathogenesis and Treatment of Bipolar Disorder (BD)"

_cells, 2025, doi:10.3390/cells14141104_

Round 1
Reviewer 1 Report
Comments and Suggestions for Authors
The tittle of this review is “Epigenetics and Gut Microbiota in the Pathogenesis and Treatment of Bipolar Disorder (BD)”, however, the majority of contents is epigenetic alteration in BD and response to pharmacotherapy. In this review, epigenetic alteration in BD is divided into three parts; DNA methylation, histone modifications and non-coding RNAs, and previous studies are summarized in Table1 and 2 for DNA methylation and non-coding RNAs. However, it is unclear what criteria the authors used to search for previous studies, and there is no information on the number of participants in each study or the platform used for the analysis.The association between pharmacotherapy for BD and epigenetic alteration is reviewed separately for valproate, lithium, and atypical antipsychotics, but the reason for the exclusion of lamotrigine is unclear.Although several studies on Gut Microbiota in BD are introduced in the latter half of the review, there is almost no information about the relationship between epigenetics and gut bacteria in BD, and the only sentence stated is "GM-derived metabolites, particularly SCFAs can function as HDAC inhibitors."
Author Response
Reviewe1
Comments and Suggestions for Authors
The tittle of this review is “Epigenetics and Gut Microbiota in the Pathogenesis and Treatment of Bipolar Disorder (BD)”, however, the majority of contents is epigenetic alteration in BD and response to pharmacotherapy. In this review, epigenetic alteration in BD is divided into three parts; DNA methylation, histone modifications and non-coding RNAs, and previous studies are summarized in Table1 and 2 for DNA methylation and non-coding RNAs.
We sincerely appreciate your valuable comments. We have carefully revised the manuscript in accordance with your suggestions. Appropriate amendments have been made, and additional information has been included, all of which are highlighted in green for your convenience.
However, it is unclear what criteria the authors used to search for previous studies, and there is no information on the number of participants in each study or the platform used for the analysis.
Author response and action taken: We added the number of participants to Table1 and 2 for DNA methylation and non-coding RNAs.
The association between pharmacotherapy for BD and epigenetic alteration is reviewed separately for valproate, lithium, and atypical antipsychotics, but the reason for the exclusion of lamotrigine is unclear.
Author response and action taken: We added a new section “3.3 Lamotrigine (LTG)”. We have modified text (highlighted in green on page 9, line 243-252)
- Although several studies on Gut Microbiota in BD are introduced in the latter half of the review, there is almost no information about the relationship between epigenetics and gut bacteria in BD, and the only sentence stated is "GM-derived metabolites, particularly SCFAs can function as HDAC inhibitors."
Author response and action taken: We added a new paragraph the relationship between epigenetics and gut bacteria in BD. We have modified text (highlighted in green on page 10, line 292-304)
Reviewer 2 Report
Comments and Suggestions for Authors
One small remark:
- The authors should perform source reference concerning with which program the figures were made (a.o. fig 1,2.)
The introduction clearly outlines what the reader can expect from the review; however, the narrative review could benefit from providing more detailed information
Author Response
Comments and Suggestions for Authors
We highly appreciate your feedback on our manuscript. We have revised the manuscript based on your valuable comments.
One small remark:
The authors should perform source reference concerning with which program the figures were made (a.o. fig 1,2.).
Author response and action taken: We added type of program that the figures were made in figure captions.
The introduction clearly outlines what the reader can expect from the review; however, the narrative review could benefit from providing more detailed information
Author response and action taken: We provided more detailed information for all sections. Please see highlights.
Reviewer 3 Report
Comments and Suggestions for Authors
The review manuscript addresses the timely topic of the interplay between epigenetics and the gut microbiota in the pathogenesis of bipolar disorder (BD). In particular, the authors review the involvement of various epigenetic mechanisms in BD and how these mechanisms may be influenced by antipsychotic drugs as well as by the gut microbiota.
The article is well-written and well-organized, providing useful information to readers on the "hot" topic of gut-brain communication.
However, for this very reason, it would be appropriate to describe, or at least mention, the variety of environmental factors known to affect the microbiota and, in turn, the gut-brain axis, such as physical activity, diet, and plant extracts.
In the final section, I would suggest mentioning and/or commenting on the possibility of modulating the microbiota through fecal microbiota transplantation.
Finally, although BDNF is mentioned, I would emphasize its role as a key mediator at the crossroads of several environmental factors that can influence the microbiota, oxidative stress, and cognitive performance (e.g., physical activity, plant extracts, learning, etc.).
Author Response
Reviewer 3
Comments and Suggestions for Authors
The review manuscript addresses the timely topic of the interplay between epigenetics and the gut microbiota in the pathogenesis of bipolar disorder (BD). In particular, the authors review the involvement of various epigenetic mechanisms in BD and how these mechanisms may be influenced by antipsychotic drugs as well as by the gut microbiota.
The article is well-written and well-organized, providing useful information to readers on the "hot" topic of gut-brain communication.
We highly appreciate your feedback on our manuscript. We have revised the manuscript based on your valuable comments.
However, for this very reason, it would be appropriate to describe, or at least mention, the variety of environmental factors known to affect the microbiota and, in turn, the gut-brain axis, such as physical activity, diet, and plant extracts.
Author response and action taken: We added a new paragraph about the variety of environmental factors known to affect the microbiota and, in turn, the gut-brain axis, such as physical activity, diet, and plant extracts in introduction. . We have modified text (highlighted in purple on page 2, line 51-63)
In the final section, I would suggest mentioning and/or commenting on the possibility of modulating the microbiota through fecal microbiota transplantation.
Author response and action taken: we added a new section “5.5. Fecal microbiota transplantation (FMT)” highlighted in purple on page 12-13, line 387-400.
Finally, although BDNF is mentioned, I would emphasize its role as a key mediator at the crossroads of several environmental factors that can influence the microbiota, oxidative stress, and cognitive performance (e.g., physical activity, plant extracts, learning, etc.).
Author response and action taken: we added more information to manuscript.
Reviewer 4 Report
Comments and Suggestions for Authors
General comments:
- Please, rephrase the title because the word “epigenetic” is adjective.
- The main topic of the review, according to the title and the introduction, is the role of microbiota in pathology of bipolar disorder through epigenetic mechanisms. Accordingly, the part of the manuscript devoted to this topic (4. The Gut Microbiome, BD and the Disease Severity) should be extended, to provide a more detailed explanations how gut bacteria can achieve such effect. Also, the discussion about the effects of sodium butyrate should be relocated from the section 3.1. Valproate, to this section, since it is one of major SCFAs produced by gut bacteria and involved in microbiota-brain interaction via epigenetic mechanisms.
- In sections 2 and 3, besides listing the genes with altered DNA methylation and non-coding RNAs that differ in BD patients compared to controls, please add some explanation regarding how these molecular changes may contribute to pathogenesis/treatment of BD.
Minor comments:
- Figure 1. can be redesigned in the same style as figures 2. and 3., for easier reading. Please, correct typos on figures 2. and 3. Also, figure 3. lacks figure legend.
- In the article Benes et al. (ref. 47) reduced expression of HDAC1 and overexpression of TGF-β and Wnt genes was shown in subjects with schizophrenia, not BD. Please, correct that.
Author Response
Reviewer 4
We sincerely appreciate your valuable comments. We have carefully revised the manuscript in accordance with your suggestions. Appropriate amendments have been made, and additional information has been included, all of which are highlighted in blue for your convenience.
Comments and Suggestions for Authors
General comments:
Please, rephrase the title because the word “epigenetic” is adjective.
Author response and action taken: we revised the title. New title: Epigenetics and Gut Microbiota in the Pathogenesis and Treatment of Bipolar Disorder (BD)
The main topic of the review, according to the title and the introduction, is the role of microbiota in pathology of bipolar disorder through epigenetic mechanisms. Accordingly, the part of the manuscript devoted to this topic (4. The Gut 92Microbiome, BD and the Disease Severity) should be extended, to provide a more detailed explanations how gut bacteria can achieve such effect. Also, the discussion about the effects of sodium butyrate should be relocated from the section 3.1. Valproate, to this section, since it is one of major SCFAs produced by gut bacteria and involved in microbiota-brain interaction via epigenetic mechanisms.
Author response and action taken: we revised the part of the manuscript and also added a new section “. Microbiota-Based Therapeutic Approaches for the Treatment of BD” that includes several sub-sections. on pages 11-14, line 287-401.
In sections 2 and 3, besides listing the genes with altered DNA methylation and non-coding RNAs that differ in BD patients compared to controls, please add some explanation regarding how these molecular changes may contribute to pathogenesis/treatment of BD.
Author response and action taken: we added more information in Table 1-2 and other sections.
Minor comments:
Figure 1. can be redesigned in the same style as figures 2. and 3., for easier reading. Please, correct typos on figures 2. and 3. Also, figure 3. lacks figure legend.
Author response and action taken: we revised all figures based on your useful comment.
In the article Benes et al. (ref. 47) reduced expression of HDAC1 and overexpression of TGF-β and Wnt genes was shown in subjects with schizophrenia, not BD. Please, correct that.
Author response and action taken: we deleted this example since the focus of article is on BD
Round 2
Reviewer 1 Report
Comments and Suggestions for Authors
The manuscript has been improved.
Reviewer 2 Report
Comments and Suggestions for Authors
no comments